# Bi-Functional Radiotheranostics of ^188^Re-Liposome-Fcy-hEGF for Radio- and Chemo-Therapy of EGFR-Overexpressing Cancer Cells

**DOI:** 10.3390/ijms22041902

**Published:** 2021-02-14

**Authors:** Yi-Shu Huang, Wei-Chuan Hsu, Chien-Hong Lin, Sheng-Nan Lo, Chu-Nian Cheng, Ming-Syuan Lin, Te-Wei Lee, Chih-Hsien Chang, Keng-Li Lan

**Affiliations:** 1Division of Isotope Application, Institute of Nuclear Energy Research, Taoyuan 32546, Taiwan; dante0319@hotmail.com (Y.-S.H.); wchsu@iner.gov.tw (W.-C.H.); chlin0805@iner.gov.tw (C.-H.L.); loshengnan@iner.gov.tw (S.-N.L.); nian.cheng@saviorlifetec.com.tw (C.-N.C.); richard5758@hotmail.com (M.-S.L.); tewei123456@gmail.com (T.-W.L.); 2Department of Biomedical Imaging and Radiological Sciences, National Yang Ming Chiao Tung University, Taipei 11221, Taiwan; 3Department of Oncology, Taipei Veterans General Hospital, Taipei 112, Taiwan; 4Institute of Traditional Medicine, School of Medicine, National Yang Ming Chiao Tung University, Taipei 11221, Taiwan

**Keywords:** cytosine deaminase, epidermal growth factor receptor, 5-fluorocuracil, liposome, prodrug, rhenium-188

## Abstract

Epidermal growth factor receptor (EGFR) specific therapeutics is of great importance in cancer treatment. Fcy-hEGF fusion protein, composed of yeast cytosine deaminase (Fcy) and human EGF (hEGF), is capable of binding to EGFR and enzymatically convert 5-fluorocytosine (5-FC) to 1000-fold toxic 5-fluorocuracil (5-FU), thereby inhibiting the growth of EGFR-expressing tumor cells. To develop EGFR-specific therapy, ^188^Re-liposome-Fcy-hEGF was constructed by insertion of Fcy-hEGF fusion protein onto the surface of liposomes encapsulating of ^188^Re. Western blotting, MALDI-TOF, column size exclusion and flow cytometry were used to confirm the conjugation and bio-activity of ^188^Re-liposome-Fcy-hEGF. Cell lines with EGFR expression were subjected to treat with ^188^Re-liposome-Fcy-hEGF/5-FC in the presence of 5-FC. The ^188^Re-liposome-Fcy-hEGF/5-FC revealed a better cytotoxic effect for cancer cells than the treatment of liposome-Fcy-hEGF/5-FC or ^188^Re-liposome-Fcy-hEGF alone. The therapeutics has radio- and chemo-toxicity simultaneously and specifically target to EGFR-expression tumor cells, thereby achieving synergistic anticancer activity.

## 1. Introduction

The development of novel nanoparticles consisting of both diagnostic and therapeutic components has increased over the past decade [1]. Liposomal nanoparticles as delivery systems own the potential for imaging and therapy of tumors. Liposome is structurally similar to cell membrane, with a hydrophobic phase within lipid bilayers and hydrophilic regions facing inward and outward. Hence, liposomal formulations serve as one of the promising approaches to encapsulate the water-soluble and water-insoluble pharmaceuticals inside the core and the lipid bilayer, respectively. Liposomes have been used widely due to their advantages such as increased absorption, delayed excretion, decreased uptake and removal from circulation by reticuloendothelial system, longer half-life within blood circulation and lower toxicity. In addition, liposomes accumulate in tumor through tumor vasculature leaky and enhanced permeability and retention (EPR) effect [2,3,4]. In addition, encapsulation of drugs within liposome results in a dramatic improvement of the pharmacokinetics, toxicity profiles and therapeutic efficacies of the drug. One of the remarkable examples is the pegylated liposomal doxorubicin, which demonstrates improved therapeutic window with less cardiotoxicity as compared with free doxorubicin and is approved for the treatment of refractory ovarian cancer and metastatic breast cancer [5,6]. Simultaneously, liposomes have also been frequently applied to the field of radiopharmaceuticals for radiotherapy and diagnostic imaging [7,8]. Preclinical anticancer studies with liposome-mediated radiopharmaceuticals, such as ^188^Re-liposome, ^99m^Tc-liposome, ^111^In-liposome, etc., have been reported [9,10]. Among of them, Rhenium-188 (^188^Re) is an ideal radionuclide for therapeutic use, due to its maximum beta emission of 2.12 MeV, with a short physical half-life of 16.9 h and its 155 keV γ emission, for imaging purposes [11], which makes ^188^Re-liposome ideal for research and clinical application [12,13,14,15].

5-Fluorocuracil (5-FU) is one of the most prescribed chemotherapy for the treatment of different cancers such as breast, bowel, skin, stomach, esophageal, pancreatic, head and neck cancers. However, it is associated with severe side effects of brain, gastrointestinal, and hematological toxicities [16,17]. A variety of strategies have been explored to curtail the severity of chemotherapy related systemic toxicity, including the use of prodrugs, biomodulation and tumor-specific drug delivery methods [18,19,20,21,22]. Systemic toxicity of 5-FU could be circumvented by introducing gene and/or antibody-directed enzyme prodrug therapy (GDEPT/ADEPT) relying on the ability of bacterial and/or yeast cytosine deaminase (CD) enzyme to convert far less toxic substrate 5-fluorocytosine (5-FC) to 5-FU and concentrating the production of 5-FU at the tumor sites [23]. Lan et. al. reported an enzyme prodrug system (Fcy-EGF/5-FC), which is a fusion protein composed yeast cytosine deaminase fused with human epidermal growth factor (Fcy-hEGF) and capable of targeting and killing EGFR-overexpressing cancer through converting 5-FC to 1000-fold toxic 5-FU. The authors showed that the viability of EGFR-overexpressing cells was significantly inhibited by Fcy–hEGF combined with 5-FC in a dose-dependent manner, and the values of IC_50_ for high EGFR-overexpressing cells were approximately 10-fold lower than those of low EGFR-overexpressing cells. This novel EGFR-targeting enzymatic prodrug system, Fcy–hEGF/5-FC, displayed potential to be a potent inhibitor for treatment of EGFR-overexpressing cancers, while reducing the severe side effects of 5-FU [24].

The past decade, several approaches to further increase the targeting and performance of PEGylated liposome, such as antibody-coated liposomes (immunoliposomes [ILs]), have been extensively studied in vitro and in vivo [25]. EGFR specific therapeutics is of great importance in cancer treatment. EGFR inhibitors such as monoclonal antibody or small molecules are either in clinical use or under development. Preclinical studies have shown that ILs bind specifically to receptors, subsequently internalizing encapsulated drugs and releasing them into the intracellular environment. Moreover, anti-EGFR ILs containing DXR (DXR-IL-C225) are under evaluation in clinical trials [26]. Nanotargeted ^188^Re-DXR-IL-C225 has been studied the cytotoxic effects for EGFR positive cancer cells in vitro [27]. Due to the EGFR-overexpressing human epithelial cancers responsible for approximately 50% of all cancer deaths, the aim of this study was to investigate the potential of a liposomal prodrug system, ^188^Re-liposome-Fcy-hEGF/5-FC, combining the cytotoxic effect of both radiotherapy and chemotherapy on EGFR-expressing cells. We conjugated the Fcy-hEGF on the surface of liposome and demonstrated the ability of conversion of 5-FC to 5-FU. Furthermore, the cytotoxic effect of ^188^Re-liposome-Fcy-hEGF/5-FC was evaluated in EGFR-overexpressing cancer cells compared with the treatment of liposome-Fcy-hEGF/5-FC or ^188^Re-liposome-Fcy-hEGF alone. We selected a potential radio- and chemo-therapeutic drug specific for EGFR-overexpressing cancers.

## 2. Results

### 2.1. Identification of Fcy and Fcy-hEGF Purified Proteins

Figure 1A represented the schematic design of Fcy-hEGF and Fcy in the pPICZ-αA vector. Fcy and Fcy-hEGF fusion proteins were prepared and purified. Briefly, either Fcy-hEGF or Fcy was cleaved by BamHI and EcoRI and ligated into the vector. Then, the host yeast strain, wild-type X-33 *P. pastoris*, was transformed and selected by Zeocin. Fcy-hEGF and Fcy proteins in the medium were bound to nickel-resin affinity column and eluted with increasing concentration of imidazole. The theoretical molecular weight of Fcy is ~17 kDa. The molecular weight of our recombinant Fcy including c-myc epitope and 6x histidine tag is ~20 kDa. In addition, the theoretical molecular weight of hEGF is ~6 kDa, thereby the molecular weight of Fcy-hEGF being ~26 kDa. Coomassie stained gels showed the band of Fcy is between 20–30 kDa and the band of Fcy-hEGF is around 30 kDa (Figure 1B). It is consistent the results of MALDI-TOF analysis, displayed the molecular weight of Fcy and Fcy-hEGF proteins are about ~20 kDa and ~26 kDa based on adjacent m/z-values, respectively (Figure 1C).

### 2.2. Preparation and Characterization of Liposome-Fcy-hEGF and -Fcy

In order to deliver the Fcy-hEGF and Fcy to the tumor efficiently, we used the liposome as the carrier. Synthesis of liposome-Fcy-hEGF and liposome-Fcy were through conjugation of proteins with maleimide-PEG_2000_-DSPE firstly by reduction using Traut’s reagent and reaction of -SH group with maleimide, followed by insertion of pegylated Fcy-hEGF or Fcy to the liposome (Figure 2A). Purification of liposome-Fcy-hEGF and liposome-Fcy was mediated by size exclusion of sepharose^TM^ 4B column to exclude the free Fcy-hEGF and Fcy proteins. The Lowry method assays showed that liposome-Fcy-hEGF and liposome-Fcy with relative large sizes flowed quickly and were collected at fractions of number 6 to 11, whereas the smaller molecules, free Fcy-hEGF and Fcy proteins, flowed slowly and were collected at fractions of number 18 to 23 (Figure 2B).

Characterization of liposome-Fcy-hEGF and liposome-Fcy was conducted by analysis of particle size, zeta potential and average amount of protein molecules/per liposome vesicle. The range of particle size and zeta potential of liposome-Fcy-hEGF and liposome-Fcy is from 94~116 nm and −20~−30 mV. The average number of Fcy-hEGF and Fcy association with liposome is 20–250 molecules from the calculation of particle size and concentration of phosphate lipid and protein based on four times of independent preparations (Figure 2C and Table 1). Flow cytometry was adopted to identify Fcy-hEGF and Fcy inserted on the liposome surface using fluorochrome-conjugated anti-myc antibody. Comparison of HNE buffer (20 mM HEPES buffer, 150 mM NaCl and 2 mM EDTA) and liposome/HNE was to obtain the pattern of liposome particles scattering in flow cytometry (Figure 2D, red circle). Results of flow cytometry showed that a significant amount of liposome-Fcy-hEGF (red line) and liposome-Fcy (green line) were stained positively by fluorochrome-conjugated anti-myc antibody as compared with the negative control of liposome (black line). Hence, Fcy-hEGF and Fcy proteins are indeed conjugated and inserted on the liposome surface (Figure 2E).

### 2.3. Cytotoxicity of Liposome-Fcy-hEGF and Liposome-Fcy

To determine whether the prodrug enzyme, Fcy-hEGF moiety, can convert 5-FC to 5-FU and specifically target EGFR-expressed cells, we performed MTT assay to evaluate the cytotoxic effect of liposome-Fcy–hEGF and liposome-Fcy in the presence of 1 mg/mL of 5-FC, on A431, MDA-MB-231 and MCF-7 cells. The expression of EGFR on cancer cell lines is A431 > MDA-MB-231 > MCF-7 [13]. Fcy alone with 5-FC did not suppress the growth of cancer cells, whereas liposome-Fcy–hEGF/5-FC and Fcy-hEGF/-5FC displayed significant inhibitory effect on the viability of A431cells compared to MDA-MB-231 and MCF-7 cells (Figure 3A). Additionally, a dose-dependent suppressive effect of liposome-Fcy–hEGF/5-FC on A431 cells was observed, whereas the inhibitory effect on MCF-7 and MDA-MB-231 reached a plateau at around 80% and 60% of cell viability, respectively. In contrast, the suppressive effects of liposome-Fcy/5-FC did not demonstrate specificity between cell lines and was less significant than the effect of liposome-Fcy–hEGF/5-FC (Figure 3A,B), indicating the critical role of hEGF of Fcy-hEGF fusion protein in targeting Fcy and 5-FU production close to EGFR-expressing cells. It also confirms the effect of selectively targeting EGFR expressing cells.

### 2.4. Cytotoxicity of ^188^Re-Liposome-Fcy-hEGF

To further enhance the cytotoxicity of liposome-Fcy-hEGF/5-FC against cancer cells, we encapsulated the radioisotope ^188^Re into the aqueous liposome interior (Figure 4A). A431 is sensitive to ^188^Re with about 20–30% inhibition of cell viability at about 100 μCi ^188^Re (Figure 4B). Further, the ^188^Re-liposome-Fcy-hEGF/5-FC with ~70% inhibition (100 μCi) revealed a better cytotoxic effect on A431 cells than the treatment of liposome-Fcy-hEGF/5-FC with ~40% inhibition or ^188^Re-liposome-Fcy-hEGF with ~40% inhibition alone as controls (Figure 4C). Therefore, we have successfully selected a potential EGFR-overexpressing cancer specific therapeutics possessing cytotoxic effect of both radioactivity and 5-FU.

## 3. Discussion

Fcy is the yeast cytosine deaminase (CD), which efficiently converts a less toxic antifungal medicine, 5-FC, into a widely administered cytotoxic drug for chemotherapy, 5-FU [28]. Hence, GDEPT/ADEPT of Fcy fusion proteins have been generated by linking Fcy to antiangiogenic protein [29], or molecules specific for tumor markers [30,31,32,33] and its microenvironment [34]. Those studies of Fcy fusion genes/proteins have shown promise in vivo by demonstrating inhibitory efficacy on a number of tumor models and xenograft systems in nude mice, in principle through the selective tumor targeting property of Fcy fusion proteins capable of converting 5-FC into 5-FU in the proximity of tumor cells. However, some factors affect the success while developing GDEPT/ADEPT of Fcy fusion proteins, such as the safety and effectiveness of a variety of carriers and vectors, gene transduction and expression efficiency, and tumor specificity [35]. Fcy-hEGF fusion protein produced by Lan et al. is the first example of linking Fcy to the endogenous ligand of receptor, such as EGFR, which is abundantly expressed by many types of cancers. It has been shown that Fcy-hEGF retained both of its ability to bind EGFR and convert 5-FC into 5-FU, with affinity and enzymatic activity similar to that of EGF and Fcy, respectively. Some potential advantages of Fcy-hEGF protein include (1) overexpression of EGFR on cancer cells providing an enormous amount of binding site for Fcy-hEGF, thereby ensuring the generation of high local concentrations of 5-FU; (2) nuclear localizing property of EGFR may enhance DNA damages resulted from 5-FU generated by Fcy-hEGF; (3) low concentration of Fcy–hEGF at nM ranges of fusion protein is sufficient for inhibiting EGFR-overexpressing cancer cells in the presence of 5-FC [24]. To further enrich the presence of Fcy–hEGF molecules within tumor site from the circulation, we used liposome as drug delivery vehicle to transfer efficiently this prodrug system, Fcy-hEGF/5-FC, into tumor microenvironment. Likewise, the EGF moiety of Fcy-hEGF will transform the passive liposome into active one specific for EGFR-overexpressing tumor. Here, we showed that Fcy-hEGF can be coupled to surface of liposomes using the insertion method (Figure 2). Liposome-Fcy-hEGF does not lose discernible extent of biological activity by conjugated process at 60 °C [36], given that the cytotoxic efficiency of liposome-Fcy-hEGF/5-FC is the same as that of Fcy-hEGF/5-FC for EGFR-overexpressing cancer cells (Figure 3).

Apoptotic cell can be induced by cytotoxic stress, such as irradiation. Several cellular factors determine the growth or arrest of cell and the type of cellular death in response to irradiation. Among cellular factors, the tumor-suppressor protein of p53 is known to be essential for apoptosis after gamma-irradiation [37,38,39,40]. Irradiation causing cell death through p53-mediated apoptosis has been confirmed by studies with p53-deficient mice that showed reduced sensitivity significantly to irradiation-induced cell death in various cell models [41,42]. This is observed in vivo and in vitro studies and is frequently observed early within 4–10 h after irradiation [43,44,45,46,47,48].

The induction of apoptosis by chemotherapy or radiotherapy is essential for the treatment of cancer. Beta-irradiation-mediated apoptosis by ^188^Re has been reported [49]. Evidences from ovarian cancer-bearing mice model treated with ^188^Re-liposome significantly increased p53 which suppressed epithelial-to-mesenchymal transition (EMT) and reversal of glycolysis. This property is essential in combating ovarian cancer [50]. We also demonstrated that the tumors from treated mice had a 26-fold increase in numbers of apoptotic cells compared with those from the normal saline control mice at 8 h after treatment with ^188^Re-liposome [12]. Excepting the mutation of p53, retinoblastoma (pRb), NOTCH, phosphoinositol-3 kinase (PI3K), phosphatase and tensin homolog (PTEN), AKT kinase, and epithelial growth factor receptor (EGFR) pathways disorder could be the potential factor to induce cell apoptosis mediated by ^188^Re-treatment [51]. ^188^Re-liposome treatment not only increased oxidative phosphorylation/glycolysis ratio but also blocked EMT [51] even restored p53 function to eliminate cancer stem cells [50]. Microarray array analysis also revealed that the tumor suppressor microRNA let-7 could be induced by ^188^Re-liposome to regulate downstream genes [52]. In addition, Chang and his colleagues also reported that ^188^Re-liposome could inhibit autophagy/mitophagy of cancer stem cells leading to tumor regression in animal models [15]. Those evidences might indicate that ^188^Re-liposome-Fcy-hEGF enhance various anti-tumor mechanism in cancer cells.

In addition to the treatment of cancer cells by beta-irradiation, the enzymatic activity of ^188^Re-liposome-Fcy-hEGF for killing cells by 5-FU did not decline after loading of ^188^Re. The tumor inhibitory efficacy of bimodality radiochemo-combination treatment of ^188^Re-liposome-Fcy-hEGF/5-FC is better than controls of single treatment, including ^188^Re-liposome-Fcy-hEGF and liposome-Fcy-hEGF/5-FC (Figure 4).

## 4. Materials and Methods

### 4.1. Reagents, Antibodies and Cell Culture

Unless otherwise specified, general reagents were obtained from Sigma-Aldrich (St. Louis, MO, USA). Sepharose^TM^ 4B was purchased from GE Healthcare (Uppsala, Sweden). Fluorochrome-conjugated anti-myc (9E10) was purchased from Zymed Laboratories (San Francisco, CA, USA). All cell culture reagents were obtained from Invitrogen (Carlsbad, CA, USA). MDA-MB-231 (stock number: BCRC 60425), MCF-7 (stock number: BCRC 60436) and A431 (stock number: BCRC 60161) were obtained from Bioresource Collection and Research Center (Hsinchu, Taiwan). MDA-MB-231 human breast cancer cells cultured in L-15 medium containing 2 mM L-glutamine and 10% heat-inactivated fetal bovine serum under an atmosphere of without CO_2_ at 37 °C. MCF-7 human breast adenocarcinoma cultured in minimum essential medium Eagle with 2 mM L-glutamine and Earle’s BSS adjusted to contain 1.5 g/L sodium bicarbonate, 0.1 mM non-essential amino acids and 1.0 mM sodium pyruvate and 10% fetal bovine serum. A431 human epidermoid carcinoma cell line cultured in Dulbecco modified Eagle medium 4 mM L-glutamine, 1 mM sodium pyruvate, 1.5 g/L sodium bicarbonate, 4.5 g/L glucose and 10% heat-inactivated fetal bovine serum under an atmosphere of 5% CO_2_ at 37 °C.

### 4.2. Expression and Protein Purification

Vectors carrying either Fcy–hEGF–myc–his_6_ or Fcy–myc–his_6_ were transformed into wild-type X-33 *P. pastoris*, plated on Zeocin-containing (200 lg/mL) agar plates and incubated for 3–4 days until the appearance of colonies. An antibody against c-myc was used to select individual colonies with high expression levels of the proteins. For large-scale expression, 0.5 L of BMD medium was inoculated with the selected colony and grown in shaker flasks to an OD600 of 8–10. Protein expression was induced with the daily addition of up to 1% methanol. Three days after induction, the protein-containing culture medium was collected and filtered before being loaded onto a nickel-resin column (Qiagen, Valencia, CA, USA) mounted on an AKTAprime plus purification system (GE Healthcare, Piscataway, NJ, USA). The column was washed with 5 mM imidazole in PBS buffer, and bound proteins were eluted with increasing concentrations of imidazole. The proteins were characterized on SDS–PAGE gels using ColorBurst™ Electrophoresis Marker (Sigma-Aldrich, C1992) as the protein marker by staining with Coomassie-blue. The protein expression of Fcy-hEGF and Fcy proteins were further identified by matrix assisted laser desorption/ionization-time of flight mass spectrometry (MALDI-TOF/MS), using acetonitril:water = 1:1 with 0.1% trifluoroacetate as the matrix solution, supplied with 10 mg/mL sinapinic acid.

### 4.3. Preparation and Identification of the Liposome-Fcy-hEGF and -Fcy

The pegylated liposome (Nano-X) (~100 nm diameter) composed of 1,2-distearoyl-sn-glycero-3-phosphocholine (DSPC), Cholesterol, and DSPE2000-PEG (molar ratio is 3:2:0.3) were prepared according to Tseng et al. [53] and was provided by Taiwan Liposome Company (Taipei, Taiwan). The storage time of Nano-X liposome is 6 months. The liposome-Fcy-hEGF and -Fcy were completed by insertion method [54]. Briefly, the Fcy-hEGF and Fcy proteins reacted with Traut’s reagent (PIERCE, Rockford, IL, USA) at molar ratio 1:5 within nitrogen condition at room temperature for 30 min. Then, the sulfated Fcy-hEGF and Fcy proteins further reacted with Mal-PEG_2000_-DSPE (NOF Corporation, Tokyo, Japan) at molar ratio 1:8 within nitrogen condition at room temperature for 2 h. The Fcy-hEGF-PEG_2000_ and Fcy-PEG_2000_ conjugates were identified by MALDI-TOF/MS (Bruker Daltonics Ultraflex III TOF/TOF) (Bruker Daltonics, GmbH, Bremen, Germany) analysis. The Fcy-hEGF-PEG_2000_ and Fcy-PEG_2000_ conjugates were purified by PD-10 column to exclude the free Mal-PEG_2000_-DSPE. To further prepare liposome-Fcy-hEGF and -Fcy, Fcy-hEGF-PEG_2000_ and Fcy-PEG_2000_ conjugates were incubated with 1 mL liposome solution at 60 °C for 30 min. The liposome-Fcy-hEGF and -Fcy were purified by sepharose^TM^ 4B to exclude the free Fcy-hEGF and Fcy proteins. A total of forty 2 mL eppendorff tubes (~1 mL/tube) were collected by gravity, and then analysis was mediated by BCA assay kit (PIERCE, Rockford, IL, USA) according to manufacturer’s protocol.

The liposome-Fcy-hEGF and -Fcy were identified by flow cytometry to confirm the Fcy-hEGF and Fcy protein molecules inserted on the surface of liposomes [55]. For flow cytometric measurements, the liposome-Fcy-hEGF and -Fcy were first blocked with 3% (*w*/*v*) fatty acid free BSA (Roche, Basel, Switzerland) for 1 h at 25 °C. After 3–4 h incubation with Fluorochrome-conjugated anti-myc antibodies on the ice, samples were measured by FACSCalibur (BD Biosciences, San Jose, CA, USA).

### 4.4. Calculation of the Amount of Fcy-hEGF and Fcy Molecules on the Surface of Liposomes

The numbers of Fcy-hEGF and Fcy molecules on each ml of liposome was determined by Lowry method, carried out using BCA kits (Thermo Scientific, Waltham, MA, USA). The particle sizes and the numbers of phospholipid molecules of each liposome was determined using Nano-ZS size analyzer (Malvern, UK) and ultraviolet-visible spectrophotometer (V-530; Jasco, Tokyo, Japan), respectively. The phospholipid content was measured using a colorimetric assay by Bartlett [56]. The amount of particle per ml of liposome solution can be calculated through the concentration of phospholipid and particle size of liposome (vesicles/mL liposome). Finally, the amount of protein molecules per vesicle through the amount of protein on liposome solution (protein molecules/mL liposome) and the amount of particle per ml liposome solution were calculated. In this experiment, the average amount of protein molecules on the surface of liposome is about 20–250, based on four independent liposome preparations.

### 4.5. Preparation of ^188^Re-Liposome-Fcy-hEGF

The ^188^W/^188^Re-generator could be homemade of Institute of Nuclear Energy Research or a commercialized GMP/Pharmaceutical-Grade ^188^W/^188^Re generator by IRE (Institut National des Radioelements, Fleurus, Belgium). Elution of the ^188^W/^188^Re generator with normal saline provided solutions of carrier-free ^188^Re as sodium perrhenate (NaReO4). The labeling method for BMEDA radiolabeled with ^188^Re was as previously described [12,13]. The labeling efficiency of ^188^Re-BMEDA complex was determined using ITLC-SG paper chromatography, eluted in normal saline. The process moved to the next step, when the labeling efficiency of ^188^Re-BMEDA complex had reached >98%. The preparation of ^188^Re-liposome-Fcy-hEGF proceeded as follows. Briefly, pegylated liposomes (1 mL, 13.5 μmol phospholipids) were added to the ^188^Re-BMEDA solution and incubated, at 60 °C, for 30 min. The effect of the ^188^Re encapsulation on the liposome size and zeta-potential has been previously studied [14]. The specification of ^188^Re-liposome for the concentration of phospholipid was 3~6 μmol/mL, for particle size was 80~100 nm, and for zeta potential was −3~2mV. The leakage of ^188^Re by in vitro stability of ^188^Re-liposome in normal saline has been previously reported [57]. The stability of ^188^Re-liposome was 93.75 ± 0.75% and 92.01 ± 1.31% (*n* = 3) at 48 h and 72 h in normal saline, respectively. The ^188^Re-liposome-Fcy-hEGF were separated from free ^188^Re-BMEDA and purified, using a PD-10 column, eluted with normal saline. In this study, the labeling efficiency of ^188^Re-liposome-Fcy-hEGF was ~50% (determined by the activity in pegylated liposomes after separation, divided by the total activity before separation).

### 4.6. MTT Assays for the Measurement of Cell Viability

A431, MDA-MB-231, and MCF-7 (5000 cells/well) cells were plated in a 96-well plate for one day. Then, cells were incubated with combinations containing 5-FC (1 mg/mL or 100 μg/mL) and different concentrations of liposome-Fcy, liposome-Fcy-hEGF, Fcy, or Fcy-hEGF in the presence or absence of ^188^Re isotope for three days. After discarding the supernatant, 200 μL of MTT solution (1 mg/mL in medium) was added to the cells for 4 h incubation at 37 °C. Then, MTT was removed and followed by addition of 200 μL of the extraction buffer (DMSO). The optical densities were measured at 570 nm by ELISA reader (Thermosystem, Multiskan Ex).

### 4.7. Data Analysis

Quantifications were based on at least three independent experiments. The data are shown as means ± the standard errors of the mean (SEM) or means ± the standard deviation (SD). The results were analyzed by one way ANOVA with Tukey’s multiple comparison method. All calculations were performed using GraphPad Prism 7 software. A *p*-value less than 0.05 was considered significant.

## 5. Conclusions

Given that approximately 50% of all cancer deaths is attributable to the EGFR-overexpressing human epithelial cancers, this led us to synthesize the ^188^Re-liposome-Fcy-hEGF/5-FC with radio- and chemo-toxicity. We demonstrated that it specifically targeted and efficiently killed ~70% of EGFR-overexpressing cancer cells. This study identifies a path towards curing EGFR-expressing cancer cells using bi-functional liposomal nanoparticles simultaneously delivering cytotoxic effects of ^188^Re radioactivity and 5-FU chemotherapy.

## Figures and Tables

**Figure 1 ijms-22-01902-f001:**
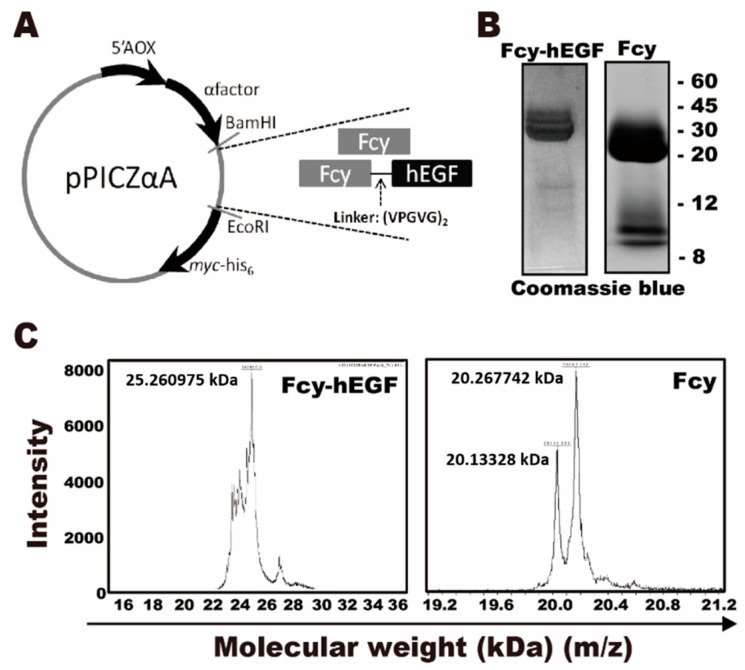
Identification of Fcy and Fcy-hEGF purified proteins. (**A**) The genes encoding Fcy–hEGF- and Fcy-myc-his_6_ were digested with the restriction enzymes, BamHI and EcoRI, and cloned into the yeast vector, pPCIZ-αA. The alpha-secreting signaling peptide is present to assist in the secretion of the proteins. (**B**) SDS-Gel results of Fcy-hEGF fusion protein (left) and Fcy purified protein (right) stained by coomassie blue. (**C**) MALDI-TOF analysis of Fcy-hEGF (left) and Fcy (right) purified proteins.

**Figure 2 ijms-22-01902-f002:**
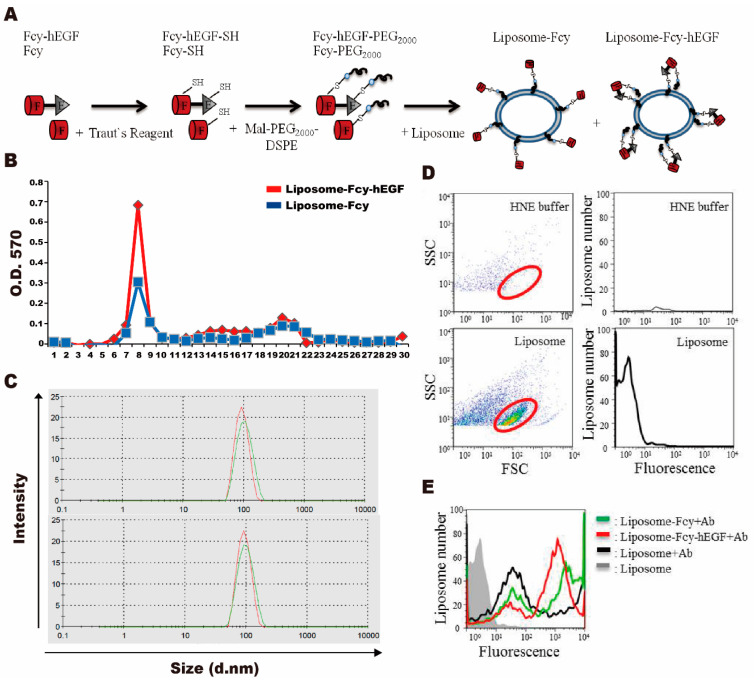
Preparation and characterization of liposome-Fcy-hEGF and -Fcy. (**A**) Fcy-hEGF and Fcy were reduced by Trout’s reagent to synthesis the thiol-conjugation of Fcy-hEGF-SH and Fcy-SH, and then conjugation with maleimide-PEG2000-DSPE to synthesis the pegylated Fcy-hEGF and Fcy, which were further inserted onto the liposome surface. (**B**) Size exclusion of liposome-Fcy-hEGF (red line) and liposome-Fcy (blue line) was mediated by sepharose^TM^ 4B column. (**C**) The size analysis of liposome (red line of top and bottom panel), liposome-Fcy-hEGF (green line of top panel) and liposome-Fcy (green line of bottom panel) by nano-ZS size analyzer. (**D**) Flow cytometric analysis of HNE buffer and liposome for gating the liposome particles (red cycle). (**E**) Flow cytometric analysis of liposome without antibody staining (gray line) was as negative control, whereas liposome (black line), liposome-Fcy-hEGF (red line) and -Fcy (green line) were stained with Fluorochrome-conjugated anti-myc antibody.

**Figure 3 ijms-22-01902-f003:**
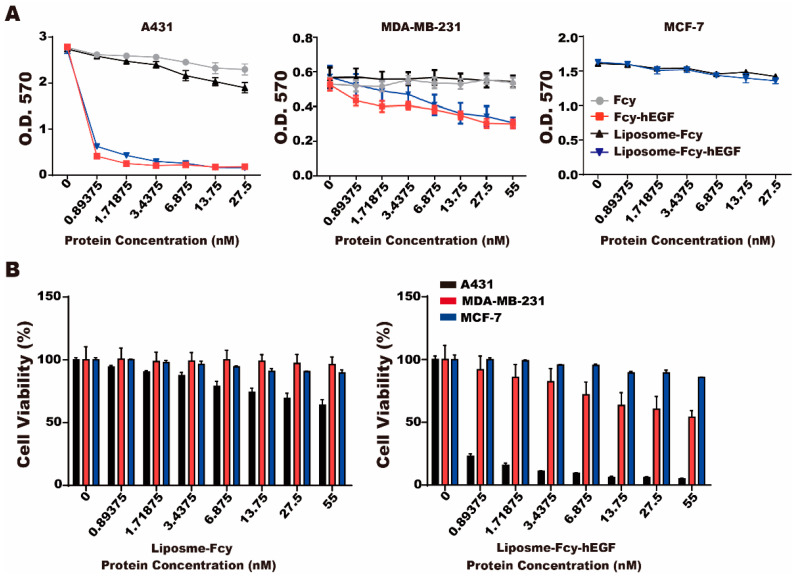
Cytotoxicity of liposome-Fcy-hEGF/5-FC and liposome-Fcy/5-FC. (**A**) MTT assay was conducted for A431 (left), MDA-MB-231 (middle) and MCF-7 (right) cells treated with different concentration of liposome-Fcy-hEGF, liposome-Fcy, Fcy-hEGF and Fcy in combination with 5-FC (1 mg/mL). (**B**) Percentage of viable A431, MDA-MB-231 and MCF-7 cells treated with different concentration of liposome-Fcy-hEGF (right) and liposome-Fcy (left) in combination with 5-FC (1 mg/mL). The data are means ± SEM of three independent experiments performed in triplicate.

**Figure 4 ijms-22-01902-f004:**
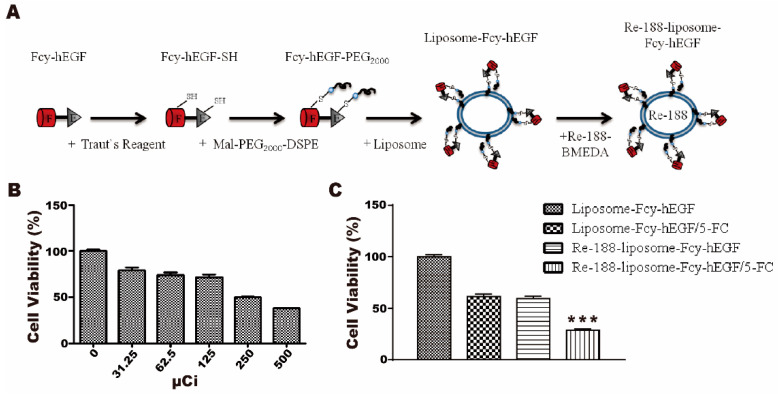
Cytotoxicity of ^188^Re-liposome-Fcy-hEGF/5-FC for A431 cells. (**A**) ^188^Re was conjugated with BMEDA first, followed by encapsulation into the interior of liposome-Fcy-hEGF. (**B**) Percentage of viable A431 cells treated with different concentration of ^188^Re. (**C**) Percentage of viable A431 cells treated with liposome-Fcy-hEGF, liposome-Fcy-hEGF/5-FC, ^188^Re-liposome-Fcy-hEGF, and ^188^Re-liposome-Fcy-hEGF/5-FC. The specific amount of ^188^Re, liposome (phospholipid), Fcy-hEGF, and 5-FC were 100 μCi, ~82 nmol, ~3.33 nM, and ~10 μg in 100 μL, respectively. The data are means ± SEM of three independent experiments performed in triplicate. *** *p* < 0.005.

**Table 1 ijms-22-01902-t001:** The average amounts of Fcy-hEGF and Fcy molecules association with liposome.

	Liposome-Fcy	Liposome-Fcy-hEGF
particle size (d. nm)	95.3 ± 1.0	107.6 ± 9.7
lipid Conc. (μmol/mL)	7.8 ± 1.6	7.7 ± 1.0
protein Conc. (μg/mL)	246.5 ± 129.5	391.4 ± 220.1
protein molecules/liposome vesicle	115.7 ± 95.8	153.6 ± 103.8
zeta potential (mV)	−22.2 ± 2.3	−26.5 ± 4.9

Data are expressed as means ± SD (*n* = 4).

## Data Availability

Not applicable.

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
