# Peer review of "Bi-Functional Radiotheranostics of 188Re-Liposome-Fcy-hEGF for Radio- and Chemo-Therapy of EGFR-Overexpressing Cancer Cells"

_ijms, 2021, doi:10.3390/ijms22041902_

Round 1

Reviewer 1 Report

The paper focuses on the development of liposomes encapsulating 188Re, functionalized with Fcy-hEGF at their surface. The approach is very interesting, and the manuscript well was written. However, some information and discussion need to be included, to improve the quality of the manuscript.

Major comments:

  1. Was the storage stability of the liposomes studied? A more detailed discussion should be included regarding the stability of the liposomes.
  2. Figure 3A: a plausible explanation for the cytotoxic behavior of Fcy–hEGF and Liposome-Fcy–hEGF observed for A431, should be described, comparing with the other cell lines tested.
  3. What was the effect of the 188Re encapsulation on the liposome size and zeta-potential?
  4. Was there any leakage of 188Re observed? Or release studies performed?
  5. How was the phospholipid quantification performed by UV-visible? Please add a more detailed procedure or include a reference. The medium and conditions for size and zeta-potential measurements are also missing.
  6. The conclusion could be improved.

Minor comments:

Page 1, line 35: Please correct “lipisomal” to “liposomal”

Page 3, Figure 1C. Please increase the quality of MALDI-TOF images and indicate the accurate value above the peaks. Moreover, all figures of the manuscript could be in better resolution. In some of them, the numbers in the graphics, are poorly perceptible.

Page 4, line 147: “…3.4 mV The average…”, a dot is missing.

Page 7, line 242: “…increased p53 increasing…” Please change one of the “increase” to not be so repetitive.

Author Response

The paper focuses on the development of liposomes encapsulating 188Re, functionalized with Fcy-hEGF at their surface. The approach is very interesting, and the manuscript well was written. However, some information and discussion need to be included, to improve the quality of the manuscript.

Major comments:

  1. Was the storage stability of the liposomes studied? A more detailed discussion should be included regarding the stability of the liposomes.

Ans: Thanks for reviewer’s comments and suggestions. The stability has been described in “Materials and Methods section. 4.3”. Preparation and identification of the Liposome-Fcy-hEGF and –Fcy. The storage stability of the liposome has been studied by Taiwan Liposome Company. Briefly, the lysolipid of Nano-X liposome should be less than 5%, and storage time of Nano-X liposome is 6 months.

  1. Figure 3A: a plausible explanation for the cytotoxic behavior of Fcy–hEGF and Liposome-Fcy–hEGF observed for A431, should be described, comparing with the other cell lines tested.

Ans: Thanks for reviewer’s comments and suggestions. This point has been clarified in “Results section. 2.3 Cytotoxicity of liposome-Fcy-hEGF and liposome-Fcy”. In our previous study by Lan et al. (2012), the expression of EGFR on cell line is A431>MDA-MB-231>MCF-7. Liposome-Fcy–hEGF/5-FC as well as Fcy–hEGF/5-FC significantly inhibits the growth of A431 cells more than MDA-MB-231 and MCF-7 cells, thereby confirming the selective action.

Ref: Lan KH, Shih YS, Chang CA, Yen SH, Lan KL. 5-Fluorocytosine combined with Fcy-hEGF fusion protein targets EGFR-expressing cancer cells. Biochem Biophys Res Commun. 2012 Nov 16;428(2):292-7.

  1. What was the effect of the 188Re encapsulation on the liposome size and zeta-potential?

Ans: Thanks for reviewer’s comments and suggestions. In our previous study by Wang et al. (2019), we have mentioned that the effect of the 188Re encapsulation on the liposome size and zeta-potential. The specification of 188Re-liposome for the concentration of phospholipid was 3~6 μmol/mL, for particle size was 80~100 nm, and for zeta potential was − 3~2mV.

Ref: Wang SJ, Huang WS, Chuang CM, Chang CH, Lee TW, Ting G, Chen MH, Chang PM, Chao TC, Teng HW, Chao Y, Chen YM, Lin TP, Chang YJ, Chen SJ, Huang YR, Lan KL. A phase 0 study of the pharmacokinetics, biodistribution, and dosimetry of 188Re-liposome in patients with metastatic tumors. EJNMMI Res. 2019 May 22;9(1):46.

  1. Was there any leakage of 188Re observed? Or release studies performed?

Ans: Thanks for reviewer’s comments and suggestions. We have studied the leakage of 188Re by in vitro stability of 188Re-liposome in normal saline by Chen et al., (2007). Briefly, we incubated 188Re-liposome in normal saline (NS) (1:1) at RT. At desired times (1, 4, 8, 12, 24, 48 and 72 h), 200 μL of 188Re-liposome solution were transferred to a column that was packed with SephadexTM G-50 (Amersham Pharmacia Biotech, Uppsala, Sweden) for the separation of 188Re-liposome from free 188Re-BMEDA complexes in normal saline. The 188Re activity was counted using a Cobra II Auto-Gamma counter (Hewlett-Packard, USA). Our results showed that the labeling stability was 93.75%±0.75% (n=3) at 48 h and 92.01%±1.31% at 72 h (n=3) in normal saline, respectively.

Ref: Chen LC, Chang CH, Yu CY, Chang YJ, Hsu WC, Ho CL, Yeh CH, Lo TY, Lee TW, Ting G (2007) Biodistribution, Pharmacokinetics and Imaging of 188Re-BMEDA-Labeled Pegylated Liposomes after Intraperitoneal Injection in a C26 Colon Carcinoma Ascites Mouse Model. Nucl Med Biol 34: 415–423

  1. How was the phospholipid quantification performed by UV-visible? Please add a more detailed procedure or include a reference. The medium and conditions for size and zeta-potential measurements are also missing.

Ans: Thanks for reviewer’s comments and suggestions. The phospholipid content was measured using a colorimetric assay by Bartlett (1959). Liposome size and zeta potential were determined by dynamic light scattering on a Nano-ZS (Malvern, UK).

Ref: Bartlett GR. Phosphorus assay in column chromatography. J Biol Chem. 1959 Mar;234(3):466-8. PMID: 13641241.

  1. The conclusion could be improved.

Ans: Thanks for reviewer’s suggestion. We have modified and improved the conclusion.

Minor comments:

Page 1, line 35: Please correct “lipisomal” to “liposomal”

Page 3, Figure 1C. Please increase the quality of MALDI-TOF images and indicate the accurate value above the peaks. Moreover, all figures of the manuscript could be in better resolution. In some of them, the numbers in the graphics, are poorly perceptible.

Page 4, line 147: “…3.4 mV The average…”, a dot is missing.

Page 7, line 242: “…increased p53 increasing…” Please change one of the “increase” to not be so repetitive.

Ans: Thanks for reviewer’s comments, we have corrected.

Reviewer 2 Report

In this manuscript, the authors constructed and characterised a bi-functional system, 188Re-liposome-Fcy-hEGF/5-FC, for radio- and chemo-therapy of EGFR-3 overexpressing cancer cells. It sounds a very interesting research topic with a good potential application. The scientific background and the ideas behind the design of the system were introduced clearly. Most of the experimental methods and results are reasonable well explained. The manuscript is scientifically sound, but several concerns need to be addressed.

  1. What is the theoretical molecular weight of the recombinant Fcy and hEGF? How they compared with the observed Mw on SDS-PAGE (Fig. 2B)? Any direct evidence for glycosylation?
  2. Lines 98-100: ‘Consequently, the results of MALDI-TOF analysis displayed many continuous peaks indicated that Fcy-hEGF and Fcy proteins were glycosylated in the yeast culture system (Figure 1C) (Figure 1C).’ - It is unclear how Fig 1C showed the protein glycosylation, the result of MALDI-TOF need be better explained. Was glycosylated peptide identified?
  3. The quality of Figure 3 is poor and hard to read, would be better to make it a colour figure. Also, it is not clearly explained what is the real difference between Fig 3 A & B. Is B calculated based on the results of MTT assay shown in A? What are the error bars stand for, SEM, n=?
  4. Lines 183-185: ‘A431 is sensitive to 188Re with about 20-30% inhibition of cell viability at about 100 μCi 188Re (Figure 4B). Further, the 188Re-liposome-Fcy-hEGF/5-FC (100 μCi, 80 nmol)…’ Then, 1 μCi/μl was mentioned in figure legend 4C. it is confusing, 100 μCi or 1 μCi/μl? Also, should state which statistical analysis was used in the figure legend.
  5. Three cell lines were tested and only A431 showed a clear positive result. Could you please discuss why and the difference between the three cell lines in terms of EGFR expression?
  6. In Method, line 287, what's the concentration of imidazole? Was it removed before next use of the proteins? Lines 288-289: ‘The molecular weight of Fcy-hEGF and Fcy proteins were further identified by matrix assisted laser desorption/ionization-time of flight mass spectrometry (MALDI-TOF/MS).’ – MALDI-TOF/MS can be used to confirm the presence of a protein, not its molecular weight. The authors need revise the statement.
  7. Method 4.4, protein concentrations were determined by Lowry method, what is reference protein used? Does it affect the real protein concentrations of Fcy-hEGF and Fcy? Why the errors for protein concentrations (Table) are so larger?  Also, ‘In this experiment, the average amount of protein molecules on the surface of liposome is about 20-250.’ - It is not very clear how it was calculated and whether the results based on three independent liposome preparations or three measurements of the same liposome preparation.

Author Response

In this manuscript, the authors constructed and characterised a bi-functional system, 188Re-liposome-Fcy-hEGF/5-FC, for radio- and chemo-therapy of EGFR-3 overexpressing cancer cells. It sounds a very interesting research topic with a good potential application. The scientific background and the ideas behind the design of the system were introduced clearly. Most of the experimental methods and results are reasonable well explained. The manuscript is scientifically sound, but several concerns need to be addressed.

  1. What is the theoretical molecular weight of the recombinant Fcy and hEGF? How they compared with the observed Mw on SDS-PAGE (Fig. 2B)? Any direct evidence for glycosylation?

Ans: Thanks for reviewer’s comments and suggestions. We did not have direct evidence to show the change of MW caused by glycosylation. We have therefore removed the description of glycosylation the text and also explained the theoretical molecular weight of our recombinant Fcy and hEGF in “Results section. 2.1. Identification of Fcy and Fcy-hEGF purified proteins”.

  1. Lines 98-100: ‘Consequently, the results of MALDI-TOF analysis displayed many continuous peaks indicated that Fcy-hEGF and Fcy proteins were glycosylated in the yeast culture system (Figure 1C) (Figure 1C).’ - It is unclear how Fig 1C showed the protein glycosylation, the result of MALDI-TOF need be better explained. Was glycosylated peptide identified?

Ans: Thanks for reviewer’s comments and suggestions. This point is related to point 1. We have therefore amended the text.

  1. The quality of Figure 3 is poor and hard to read, would be better to make it a colour figure. Also, it is not clearly explained what is the real difference between Fig 3 A & B. Is B calculated based on the results of MTT assay shown in A? What are the error bars stand for, SEM, n=?

Ans: Thanks for reviewer’s comments and suggestions. The Figure 3 has been made with color figure and re-drafted using Adobe graphics software to improve their quality. Figure 3A showed the comparison of each drug to individual cell line and Figure 3B showed the comparison of three cell lines sensitive to Liposome-Fcy (left panel) and to Liposome-Fcy-hEGF (right panel). Yes, Fig 3B was calculated based on the results of MTT assay shown in Fig. 3A. The error bars are the mean ± SEM (n=3), which has now been made clearer in the Figure Legend.

  1. Lines 183-185: ‘A431 is sensitive to 188Re with about 20-30% inhibition of cell viability at about 100 μCi 188Re (Figure 4B). Further, the 188Re-liposome-Fcy-hEGF/5-FC (100 μCi, 80 nmol)…’ Then, 1 μCi/μl was mentioned in figure legend 4C. it is confusing, 100 μCi or 1 μCi/μl? Also, should state which statistical analysis was used in the figure legend.

Ans: Thanks for reviewer’s comments and suggestions. The radioactivity of 188Re-liposome-Fcy-hEGF/5-FC group was 100 μCi in Figure 4C. The statistical analysis for Figure 4C is one-way ANOVA with Tukey’s multiple comparison. The description has now been made clearer in the Figure Legend. The activity concentration is 1 µCi/µl that we prepared for experiments.

  1. Three cell lines were tested and only A431 showed a clear positive result. Could you please discuss why and the difference between the three cell lines in terms of EGFR expression?

Ans: Thanks for reviewer’s comments and suggestions. This point has been clarified in “Results section. 2.3 Cytotoxicity of liposome-Fcy-hEGF and liposome-Fcy”. In our previous study by Lan et al. (2012), the expression of EGFR on cell line is A431>MDA-MB-231>MCF-7. Liposome-Fcy–hEGF/5-FC as well as Fcy–hEGF/5-FC significantly inhibits the growth of A431 cells more than MDA-MB-231 and MCF-7 cells, thereby confirming the selective action.

Ref: Lan KH, Shih YS, Chang CA, Yen SH, Lan KL. 5-Fluorocytosine combined with Fcy-hEGF fusion protein targets EGFR-expressing cancer cells. Biochem Biophys Res Commun. 2012 Nov 16;428(2):292-7.

  1. In Method, line 287, what's the concentration of imidazole? Was it removed before next use of the proteins? Lines 288-289: ‘The molecular weight of Fcy-hEGF and Fcy proteins were further identified by matrix assisted laser desorption/ionization-time of flight mass spectrometry (MALDI-TOF/MS).’ – MALDI-TOF/MS can be used to confirm the presence of a protein, not its molecular weight. The authors need revise the statement.

Ans: Thanks for reviewer’s comments and suggestions. We have revised the statement in “Materials and Methods 4.2”. The imidazole we used was 5 mM and would be removed when the Fcy-hEGF-PEG2000 and Fcy-PEG2000 conjugates were purified by PD-10 column before next use of protein. The Fcy-hEGF and Fcy proteins solution were mixed with a saturated sinapinic acid matrix solution at a volumetric ratio (Vsample/Vmatrix) of 1:10. The solution (5 μl) was then transferred to the MALDI target surface. After the solution was dried under air, the target was induced into the MALDI ion source of a MALDI-TOF/TOF mass spectrometer (Bruker Daltonics Ultraflex III TOF/TOF) (Bruker Daltonics, GmbH, Bremen, Germany) operated in linear mode. The sample spots were irradiated with a pulsed nitrogen laser (337 nm) for desorption and ionization. In this study, the MALDI-TOF/TOF mass spectrometer could be used to confirm the presence of a Fcy-hEGF and Fcy proteins. We have used protein expression instead of molecular weight.

  1. Method 4.4, protein concentrations were determined by Lowry method, what is reference protein used? Does it affect the real protein concentrations of Fcy-hEGF and Fcy? Why the errors for protein concentrations (Table) are so larger?  Also, ‘In this experiment, the average amount of protein molecules on the surface of liposome is about 20-250.’ - It is not very clear how it was calculated and whether the results based on three independent liposome preparations or three measurements of the same liposome preparation.

Ans: Thanks for reviewer’s comments and suggestions. Lowry method were carried out using BCA kits (Thermo scientific, USA) using bovine serum albumin (BSA) as reference protein to determine the coupling efficiency, which is a common and stander method for protein quantification. The errors for protein concentration so huge are due to calculation based on four times of independent liposome preparations. Each preparation had different conjugation efficiency. The surface of liposome was calculated based on four times of independent liposome preparations, which has now been made clearer in “Materials and Methods 4.4 Calculation of the amount of Fcy-hEGF and Fcy molecules on the surface of liposomes”.

Round 2

Reviewer 1 Report

All questions and suggestions were performed by the authors. However, the answers given by the authors to the following questions should be included in the manuscript:

3. What was the effect of the 188Re encapsulation on the liposome size and zeta-potential?

4. Was there any leakage of 188Re observed? Or release studies performed?

5. How was the phospholipid quantification performed by UV-visible? lease add a more detailed procedure or include a reference. The medium and conditions for size and zeta-potential measurements are also missing.

Author Response

Reviewer 1: Comments and Suggestions for Authors

All questions and suggestions were performed by the authors. However, the answers given by the authors to the following questions should be included in the manuscript:

3. What was the effect of the 188Re encapsulation on the liposome size and zeta-potential?

Ans: Thanks for reviewer’s comments and suggestions. We have amended the text with red color in 4.5. Preparation of 188Re-liposome-Fcy-hEGF. The effect of the 188Re encapsulation on the liposome size and zeta-potential has been previously studied [14]. The specification of 188Re-liposome for the concentration of phospholipid was 3~6 μmol/mL, for particle size was 80~100 nm, and for zeta potential was − 3~2mV.

4. Was there any leakage of 188Re observed? Or release studies performed?

Ans: Thanks for reviewer’s comments and suggestions. We have amended the text with red color in 4.5. Preparation of 188Re-liposome-Fcy-hEGF. The leakage of 188Re by in vitro stability of 188Re-liposome in normal saline has been previously reported [58]. The stability of 188Re-liposome was 93.75% ± 0.75% and 92.01% ± 1.31% (n=3) at 48 h and 72 h in normal saline, respectively.

5. How was the phospholipid quantification performed by UV-visible? lease add a more detailed procedure or include a reference. The medium and conditions for size and zeta-potential measurements are also missing.

Ans: Thanks for reviewer’s comments and suggestions. We have amended the text with red color in 4.4. Calculation of the amount of Fcy-hEGF and Fcy molecules on the surface of liposomes. The phospholipid content was measured using a colorimetric assay by Bartlett [57].

Reviewer 2 Report

The revised manuscript has address all the question sufficiently, but a minor correction or clarification needed. 

Lines 261-262, 'The column was washed with 5 mM imidazole in PBS buffer, and the proteins were eluted by the same buffer.' How can the proteins were washed and eluted with the same buffer? Was imidazole concentration increased?

Author Response

Reviewer 2: Comments and Suggestions for Authors

The revised manuscript has address all the question sufficiently, but a minor correction or clarification needed. 

Lines 261-262, 'The column was washed with 5 mM imidazole in PBS buffer, and the proteins were eluted by the same buffer.' How can the proteins be washed and eluted with the same buffer? Was imidazole concentration increased?

Ans: Thanks for reviewer’s comments and suggestions. We have amended the text in 4.2. Expression and protein purification. The column was washed with 5 mM imidazole in PBS buffer, and bound proteins were eluted with increasing concentrations of imidazole using the AKTAprime plus purification system.